# Internalized Sexual Stigma and Mental Health Outcomes for Gay, Lesbian, and Bisexual Asian Americans: The Moderating Role of Guilt and Shame

**DOI:** 10.3390/ijerph21040384

**Published:** 2024-03-22

**Authors:** Kian Jin Tan, Joel R. Anderson

**Affiliations:** 1School of Behavioural and Health Sciences, Australian Catholic University, Melbourne Campus (St Patrick), Locked Bag 4115, Melbourne, VIC 3065, Australia; kianjin.tan@latrobe.edu.au; 2Australian Research Centre in Sex, Health and Society (ARCSHS), La Trobe University, Melbourne, VIC 3086, Australia

**Keywords:** internalized sexual stigma, mental health, shame, guilt, Asian American, lesbian, gay bisexual

## Abstract

The literature unequivocally demonstrates that lesbian, gay, and bisexual (LGB) individuals experience disproportionate mental health and social wellbeing impacts. Here, we respond to recent calls for research in the field of sexual minority health to better understand why various overlapping and intersecting identities can further drive health disparities. In this paper, we focus on the specific intersections of ethnicity and sexuality for Asian LGB individuals and the role of internalized stigma in driving poorer mental health outcomes for this group. We recruited 148 LGB Asian participants residing in the United States (*M_age_* = 22.82 years, *SD* = 4.88) to participate in our online cross-sectional survey in which we collected data on their internalized stigma, levels of guilt and shame about their sexuality, and measures of depression, anxiety, and distress. Contrary to our predictions, there were no bivariate relationships between internalized sexual stigma and any of the mental health outcomes. However, a parallel mediation analysis revealed that guilt, but not shame, mediates the relationship between internalized sexual stigma and all mental health outcomes (depression, anxiety, and stress) for LGB Asian American individuals. This research highlights the important of exploring additional variables that may exacerbate of protect against poor mental health for individuals with multiple intersecting identities.

## 1. Introduction

### 1.1. Minority Stress Theory and Internalized Sexual Stigma

LGB adults and youths experience higher rates of mental health disorders, substance use disorders, and suicidality compared to cisgender heterosexuals [1,2,3,4]. For instance, a recent Australian survey reported that 16% of same-sex-attracted individuals had attempted suicide compared to 3% of heterosexual individuals [5]. One explanation for these health disparities comes from minority stress theory [6,7], which posits that systemic power imbalances against marginalized groups, such as LGB people in a heteronormative society, could lead to its members to experience unique minority stress processes that directly and negatively impact their mental health. The three minority stress processes posited by this theory are internalized sexual stigma ISS: also known colloquially as internalized homophobia or heterosexism, [8], subjectively perceived stigma, and objective experiences of prejudice events.

In their seminal study on ISS, Meyer hypothesized that each of these three minority stress processes would have independent effects on psychological distress. This hypothesis was tested in a sample of 741 ‘out’ gay men in New York City. At the bivariate level, each of the three processes were correlated to measures of psychological distress (i.e., demoralization, guilt, suicidality, AIDS-related traumatic stress response, and sexual dysfunction); however, ISS was the only one of these processes that significantly predicted all measures of psychological distress in a multiple regression [7]. This suggests that by understanding the societal and individual mechanisms that impact ISS, the internal and external structures that lead to health disparity experienced by the LGB community could be further elucidated. Since then, a range of studies have continued along this line of research into social determinants and relevant health outcomes in other sexuality-specific populations. For example, studies have shown that ISS is related to increased experiences of somatization, obsessive compulsiveness, interpersonal sensitivity, depression, and anxiety symptomology for lesbian women [9], identity concealment for bisexual men [10], and binge eating in lesbian and bisexual women [11]. Minority stress theory is arguably the most commonly used theoretical framework to understand health and wellbeing concerns for LGB individuals, and the evidence clearly shows that ISS is a force in driving LGB–heterosexual disparities.

There is a growing body of research demonstrating that the relationship between ISS and psychosocial outcomes is modulated by a range of demographic and psychosocial factors; see [12] for a recent review of longitudinal evidence. These include access to social supports [13,14], identity disclosure [15], relationship with religion [16], identity factors [17], and access to HIV/STI testing and information [18]—more specifically, a lack of access to social support, decreased identity disclosure, increased conflict between religious and sexual identities, and a lack of access to HIV/STI testing and information exacerbates the ISS–outcomes relationship. The specific focus of this paper is the intersection of sexuality and ethnicity.

### 1.2. ISS at the Intersection of Sexuality and Ethnicity

*Intersectionality* stems from Black feminism and *critical race theory* to address the systemic oppression and social inequities experienced by multiple oppressed minority identities [19]. For example, Asian women do not experience racism in the same as Asian men due to their gender, and they do not experience sexism the same as White women due to their race. As such, by diminishing and erasing these unique intersectional experiences of prejudice of multiple marginalized individuals, the systemic discrimination and oppression against them can persist [19]. Psychology research has also highlighted the need to adopt an intersectional framework to highlight between and within group effects, such as for LGB people of color [20,21,22].

Although not driven by intersectionality, Meyer [7] proposed that LGB individuals who are members of a marginalized ethnic group experience substantial social and cultural minority stressors due to their ethnicity in addition to sexual minority stress. e.g., [23,24]. While the majority of research in this space has been conducted in Western countries, resulting in data from Christian-centric and White-majority samples, there is a small amount of research exploring ISS in samples with underrepresented ethnicities that substantiates Meyer’s claim. For example, there have been positive correlations reported between ISS in LGB people of color and ethnic victimization, racism, and internalized racism [24,25,26]. The recent body of evidence also supports the claims that LGB people of color have comparatively higher levels of ISS than White LGB individuals in Western countries, e.g., [24,27,28].

One key outcome of the literature exploring ISS for LGB people of color has been the identification of several unique stressors that they experience, which White LGB individuals do not. For example, sexual stigma from their ethnic community [29], racism from the LGB community [30], and general unfair treatment due to their ethnicity [31]. These unique stressors may explain the poorer mental health outcomes and higher rates of suicidality that LGB people of color report compared to White LGB individuals [4,25,32]. Taken together, there are marginalized subgroups within the LGB community, including ethnicity, and there are a range of factors that differentially impact levels of ISS across LGB ethnic groups (due to their multiple marginalized identities).

### 1.3. Internalized Sexual Stigma for Lesbian, Gay, and Bisexual Asians

Different LGB ethnic groups might have culture-specific factors that differentially impact ISS levels. For instance, Asian cultures have a range of values and beliefs that stem from their ethnohistorical backgrounds that may affect levels of ISS in LGB Asian people. Some of these unique Asian cultural values and beliefs include national criminal legalization towards sexual minorities [33], family honor [34], filial piety [35,36], collectivism [37], and ethnic self-identification [38]. Supporting this claim, our recent systematic literature review and meta-analysis by [28] of 28 studies revealed higher levels of ISS in LGB Asian individuals compared to LGB White individuals (moderate effect size: Hedges’ *g* = 0.52). This review also revealed that some of these Asian-specific values and beliefs were associated with higher levels of ISS [26,37,39]. This suggests that the intersectional sexual and ethnic minority stress that LGB Asians uniquely experience may exacerbate their levels of ISS.

In addition to experiencing elevated levels of ISS, LGB Asian individuals may also experience feelings of shame and guilt about their sexual identity in a way that is different to non-Asian individuals. According to Lewis [40], one experiences shame when the self is the target of negative evaluations, while in contrast, guilt arises when one’s behavior is the focus. Although an action or transgression can engender both feelings, shame in particular is often viewed as a reflection of an underlying sense of worthlessness or defectiveness [16,40]. In samples of gay men, guilt and shame have been reported to be independently related to poorer mental health [41], a failed ability to integrate sexuality with religion [42], and it mediates the relationship between intrapersonal identity stresses and binge eating [11,43].

Confucianism values and beliefs are dominant in many East and South-East Asian cultures. These values portray mainly patriarchal, heteronormative, and collectivistic sociopolitical views on what makes a good person [44]. For example, these values specify the importance of one’s role in the family—specifically, that family should be considered in heterosexual terms, with gender normative roles, and that one’s role in the family is an important feature in defining one’s self-worth and position in the community [44,45]. In addition to this expectation around heteronormativity, there is evidence to suggest that non-heterosexuality should be concealed in order to protect their family honor. For instance, Rehman and colleagues [46] conducted a qualitative thematic analysis in the UK to explore the minority stress experienced by Black, Asian, and Minority Ethnic (BAME) LGB individuals. The South Asian community members expressed that they feel the burden of not damaging their family’s “izzat” (honor/respect) that leads to shame within the community and increasing the risk of being the targets of gossip and ridicule [46]. These values of Confucianism and family honor are associated with both ISS, e.g., [35,36,44] and also with guilt and shame, e.g., [45,46,47]. The second major aim of this study was to explore shame and guilt as mediating factors of the relationship between ISS and mental health.

### 1.4. Aims and Hypotheses

The literature exploring intersectional identities within the LGBTQ+ community is growing, and there is a continued need to expand the evidence base from samples differing ethnic identities. As such, there is an overgeneralizing of understandings of minority stress for underrepresented ethnic identities from the evidence for minority stress in White LGB individuals. In this paper, we have two major aims. Our first major aim is to contribute to this limited literature by exploring ISS and its relationship to mental health in a sample of LGB Asian Americans. As argued above, the unique Asian sociocultural view on sexuality provides a novel intersection of ethnic and sexual identities. As such, our second major aim is to explore the roles of guilt and shame in explaining the relationship between ISS and mental health for Asian Americans, who have these intersecting identities. Based on the current literature and theoretical frameworks, the following hypotheses were formulated:

**H1****.** 
*The minority stress hypothesis.*


Based on the premise of minority stress theory [6,7] and evidence suggesting that intersecting ethnic and sexual identities can drive feelings of marginalization and conflict, we predicted that ISS will be positively correlated with self-reported mental health (i.e., symptomology scores on measures of depression, anxiety, and stress).

**H2****.** 
*The shame and guilt hypothesis.*


Based on evidence that shame and guilt appear to mediate the relationship between ISS and other self-relevant outcomes (e.g., intrapersonal identity stresses and binge eating; [11,43]), we predicted that shame (**H2a**) and guilt (**H2b**) would mediate the ISS–mental health relationships in H1.

## 2. Methods

### 2.1. Study Design

This study employed an online, cross-sectional survey design to collect data on ISS (independent variables), guilt and shame (mediating variables), and depression, anxiety, and stress (dependent variables).

### 2.2. Participant Characteristics

An a priori power analysis was conducted in G*Power (version 3) to determine the minimum sample size needed to detect an effect size of *f*^2^ = 0.10 (α = .05, Power = .80) for a linear multiple regression design with three test predictors (i.e., ISS, shame, and guilt). The analysis determined that 114 Asian American participants would be needed to test the hypotheses of the study. An advertisement was displayed to the target sample group on Prolific^TM^, a crowdsourcing platform that allows for targeted recruitment. The inclusion criteria were that participants must be non-heterosexual Asian people of at least 18 years of age who reside in the United States (participants not meeting these criteria were not given access to the advertisement, and thus could not participate). In response, we recruited 148 participants (*M_age_* = 22.82 years, *SD* = 4.88) who completed the survey.

The sample were mostly cisgender (*n* = 135 [91.2%], 75 men [50.7%] and 60 women [40.5%]), and a minority of the participants identified as transgender, non-binary, gender diverse, or with multiple genders (*n* = 13 [8.8%]). In regard to their sexuality, participants identified as bisexual (*n* = 78 [52.7%]), gay or lesbian (*n* = 60 [40.5%]), and a minority identified as either asexual, queer, or self-identified (*n* = 10 [6.8%]). The majority were atheists or agnostic (*n* = 95 [64.2%]). The largest religious group was Christian, including Catholics (*n* = 23 [15.5%]), followed by Buddhist (*n* = 15 (10.1%]), Hindu (*n* = 12 [8.1%]), Muslim (*n* = 6 [4.1%]), spiritual (*n* = 6 [4.1%]), Jewish (*n* = 1 [0.6%]), and other religious affiliation (*n* = 4 [2.7%]). Participants were able to select more than one response to the question about their religious identification, and thus the responses total more than the number of participants in the sample. In regard to relationship status, the majority were single (*n* = 92 [62.2%]) and the remaining were in a relationship with one or more partners (*n* = 56 [37.8%]).

### 2.3. Measures

#### 2.3.1. Predictor Variables

The *Internalized Sexual Prejudice* scale (ISP; [48]) was used to measure participant levels of sexual stigma towards their sexuality by endorsing 15 items (e.g., ‘I sometimes resent my sexual orientation’) on a scale ranging from 1 (*strongly disagree*) to 5 (*strongly agree*). Appropriate items were reversed scored and averaged, where higher scores indicate greater levels of ISS. There is evidence for convergent validity with depression (β = 0.32, *p* < 0.05) and self-esteem (β = 0.37, *p* < 0.001) and divergent validity with collective action (β = 0.23, *p* > 0.05; [48]). This scale yielded adequate levels of internal reliability across sexuality in this sample (Cronbach’s α = .91).

Single items for guilt and shame were used to measure participants levels of guilt and shame in association with their sexuality, ‘To what extent do you feel guilty [ashamed] about being gay [lesbian/bisexual]?’ [49]. Participants responses on a scale ranging from 1 (*strongly disagree*) to 5 (*strongly agree*), with higher scores indicating higher degrees of guilt and shame. Participants’ intuitive understanding of these experiences were relied upon without specific definition of these affective experiences.

#### 2.3.2. Mental Health Outcome Variables

The short form of the *Centre for Epidemiologic Studies Depression* (CESD-10; [50]) scale was used to measure participants’ levels of depressive symptomology in the past week by endorsing 10 items: three items on depressive affect, five items on somatic symptoms, and two items on positive affect. The scale ranged from 0 (*rarely or none of the time* [*less than 1 day*]) to 3 (*all of the time* [*5–7 days*]). The positive affect items were reverse scored and averaged, where higher scores indicate greater severity of depressive symptomology. A cut-off score (when the items scores are summed) of 10 was used to distinguish participants classified as ‘clinically depressed’ [50,51,52]. The CESD-10 has established convergent validity with self-reported health (*r* = 0.37) and positive affect (*r* = −0.63; [50]). It also has moderate consistency over 3 years in a Chinese elderly population (*r* = 0.44, *p* < 0.01; [52]). This scale yielded adequate levels of internal reliability in this sample (Cronbach’s α = 0.88).

The *General Anxiety Disorder* (GAD-7; [53]) scale was used to measure participants’ levels of anxiety symptoms in the last two weeks by endorsing seven items (e.g., ‘Trouble relaxing’) on a scale ranging from 0 (*not at all*) to 3 (*nearly every day*). The scores were averaged, where higher scores indicate greater severity of anxiety symptoms. A cut-off score (when the items scores are summed) of 10 was used to distinguish participants classified as potentially having an anxiety disorder. The GAD-7 has established concurrent validity with the Beck Anxiety Inventory (*r* = 0.72) and the anxiety subscale of the Symptom Checklist-90 (*r* = 0.74), and has good test-retest reliability (intraclass correlation = 0.83; [54]). This scale yielded adequate levels of internal reliability in this sample (Cronbach’s α = .93).

Finally, the *Perceived Stress Scale* (PSS; [55]) was used to measure participants’ rates of perceived stress in the last month by endorsing 10 items (e.g., ‘In the last month, how often have you felt nervous or “stressed”?’) on a scale ranging from 0 (*never*) to 4 (*very often*). Appropriate items were reversed scored and averaged, where higher scores indicate greater levels of stress. The PSS has established concurrent validity with the State-Trait Anxiety Inventory–Trait version (*r* = 0.73), and divergent validity with Sensation-Seeking Scale (*r* = −0.04) and the Santa Clara Strength of Religious Faith Questionnaire–Short Form (*r* = 0.02; [56]). It also has good test-retest reliability in a college sample after 2 days (*r* = 0.85) and in a community sample after 6 weeks (*r* = 0.55; [55]). This scale yielded adequate levels of internal reliability in this sample (Cronbach’s α = .88).

### 2.4. Data Analysis

Data were analyzed using parametric analyses in IBM SPSS Statistics version 27. To investigate if ISS was correlated with any of the mental health outcomes (H1), a bivariate correlation analysis was conducted. The effect sizes of the Pearson’s *r* were interpreted based on the following guidelines: 0.1 = small, 0.3 = medium, and 0.5 = large [57]. Second, a series of regressions were conducted in order to explore the residuals as part of assumption testing for the mediation analyses. Finally, parallel mediation analyses were conducted using PROCESS v4.0 [58] to 5000 bootstrapped samples to investigate if the relationships between ISS and the mental health outcomes were mediated by guilt and shame (H2). The significance of the mediation pathways was based on the 95% confidence intervals not crossing 0 (*p* < 0.05), and we note that PROCESS uses the bootstrapped method of mediation analysis (as is standard practice, see [59]).

### 2.5. Procedure

Participants responded to an advertisement that was placed in Prolific^TM^ to complete an online survey on sexuality and the lesbian, gay, bisexual, transgender, queer/questioning, and other (LGBTQ+) community created with Qualtrics (reviewed and approved as a low risk protocol by the Human Research Ethics Committee of Australian Catholic University (HREC2021-83EAP—approved December 2021). After reading the information letter, those who agreed to participate were redirected to the website hosting the survey (http://www.qualtrics.com/, accessed in August of 2022). Participants completed the demographic questions, then the 12 measures in a randomized order (to limit presentation effects). Participants were then debriefed, thanked for their time, and asked if they agree for their anonymized data to be included in this study. Participants were reimbursed $2 in exchange for their time.

## 3. Results

### 3.1. Data Preparation and Screening

There were no excessive scores identified on the ISS and mental health variables (i.e., all *z*-scores were within the recommended criterion of *z* > |3.29|; [57]); however, there was one participant with guilt and shame scores that appeared to be outliers (*z*_guilt_ = 3.49; *z*_shame_ = 3.98). The participant’s guilt and shame scores were treated by a process of winsorizing (i.e., replacing excessive scores with *M* ± 3*SD*; [57]). Visual inspection of the histograms, scatterplots, and *P*-*P* plots revealed no violations of linearity, normality, or homoscedasticity. There were no cases missing from the data set.

### 3.2. Descriptive Statistics and Univariable Analysis

Table 1 presents the descriptive statistics and bivariate correlations between ISS, guilt, shame, and mental health outcome variables. On average, the participants had average levels of ISS with moderately low levels of self-reported shame and guilt and poor levels of mental health. Against predictions (H1), ISS did not correlate with any mental health outcomes, although it was strongly correlated with guilt and shame. Guilt, but not shame, had small positive correlations with all three mental health outcomes. Guilt and shame were strongly related to each other.

### 3.3. Mental Health Classifications

Using the CESD-10 and GAD-7 cut-offs, 70.9% and 48.0% of participants could be classified as ‘clinically depressed’ (*n* = 105) and having an anxiety disorder (*n* = 71), respectively (no cut-off scores were available for the measure of stress).

### 3.4. Mediation Analyses

Prior to interpreting the results of the mediation analyses, several assumption tests for hierarchical multiple regression analyses were conducted. The assumption of independent errors was met (Criteria = 1–3; D-W_observed_ = 2.24–2.43). Cook’s distances fell within the acceptable range (d_observed_ ≤ 0.21), and Mahalanobis’ distances (d_observed_ ≤ 14.36) did not exceed the critical value χ^2^(3) = 16.27, *p* < 0.001, suggesting that multivariate outliers were not an issue. The collinearity of univariate dependent variables was also assessed and was found to be acceptable (1.74 ≤ VIF ≥ 2.53).

To test whether the relationship between ISS and the mental health outcomes was mediated by guilt and shame, we entered ISS as the predictor and guilt and shame as parallel mediators across three mediation models in which depression, anxiety, and stress served as the outcome variables. In contrast to our predictions for the mediating role of shame (H2a), the indirect effect through shame was not significant for depression (*B* = −0.05, 95% CI [−0.25, 0.15]; see Figure 1), anxiety (*B* = −0.02, 95% CI [−0.20, 0.15]; see Figure 2), or stress (*B* = −0.03, 95% CI [−0.23, 0.13]; see Figure 3). However, supporting our predictions for the mediating role of guilt (H2b), the indirect effect through guilt was significant for depression (*B* = 0.18, 95% CI [0.06, 0.32]; see Figure 1), anxiety (*B* = 0.13, 95% CI [0.02, 0.26]; see Figure 2), and stress (*B* = 0.15, 95% CI [0.03, 0.28]; see Figure 3).

## 4. Discussion

This study aimed to explore the relationship between ISS and mental health at the intersection of sexuality and ethnicity in Asian American LGB individuals. In addition, we explored the mediating roles of guilt and shame in explaining this relationship. Contrary to our predictions, the minority stress hypothesis (H1) was not supported—that is, there were no relationships between ISS and depression, anxiety, or stress. A parallel mediation analysis provided partial support for the shame and guilt hypotheses (H2). Specifically, the shame hypothesis (H2a) was not supported for any of the three mental health outcomes. However, the mediation analyses provided evidence supporting the guilt hypothesis (H2b)—that is, guilt mediates the relationship between ISS and all mental health outcomes (depression, anxiety, and guilt) for LGB Asian individuals.

### 4.1. Discussion of Major Findings

Of note, one of the starkest findings of the analysis was that our sample of Asian LGB individuals reported concerningly high rates of mental health symptomology. Specifically, the sample reported substantially higher rates of depression (70.9%) and anxiety (48.0%) than the general American population for who the rates are 18.5% [60] and 15.6% [61], respectively. This is cohesive with the application of an intersectional lens to minority stress theory, whereby identifying as both Asian and LGB (i.e., multiple marginalized identities) leads to a comparatively higher prevalence of mental health issues [4,25,32]. However, and against our expectations, ISS did not directly relate to (at the bivariate level) depression, anxiety, or stress. This is contrary to predictions based in theories and to previous studies that reported associations between ISS with a range of health outcomes, including depression and anxiety (e.g., [7,9,34]). Although not frequent, there have been other studies that have reported no significant correlations between ISS and mental health outcomes in certain subgroups within the LGB community. For example, ISS was correlated with neither depression or self-esteem in lesbians [62], nor depressive symptoms or suicidal ideation in bisexual women [63]. These rather contradictory results may be due to protective factors that can conceal the relationship, such as strength-based factors (e.g., resilience) and identity integration (see [42]).

Perrin and colleagues [64] conducted a path analysis to identify strength-based factors that may influence the mental health and positive health behaviors of LGB and gender diverse individuals. From their study, they reported that social support, community connectedness, identity pride, self-esteem, and resilience all positively influence mental health and positive health behaviors either directly or indirectly [64]. The other possibility is that Asian LGB individuals are integrating their sexual and ethnic identities as an identity management strategy. *Identity integration* is a process of restoring the threatened social identities (in this case Asian and LGB) by blending and reconciling them to reduce the negative impacts of having conflicting identities [65]. This has been used as a framework to understand how individuals with perceived conflicting identities may not experience negative mental health and wellbeing outcomes. There has been a growing body of evidence that proposes that identity integration could act as a protective factor in a range of intersecting identities within the LGB community. This includes the intersections of sexuality with gender [66], religion [49], and professional identities [67]. Further studies with foci on the impact of strength-based factors and identity integration on the relationship between ISS and mental health are suggested, in particular at the intersection of sexuality and ethnicity.

Another unexpected finding was that shame did not significantly correlate nor mediate the relationship between ISS and any mental health outcomes, yet guilt did. These relationships may be partially explained by the locus of the object of negative evaluation. The item that was used to measure the personal experience of shame in association with the participant’s sexuality was, “To what extent do you feel ashamed about being gay [lesbian/bisexual]?”, which places the self as the object of negative evaluation. This is coherent with the definition of shame by Lewis [40]. ISS is also an internal form of stress that directs societal negative values and evaluations towards oneself [68]. Comparatively, guilt arises from negative evaluations towards one’s behaviors [40]. Gilbert and colleagues [68] reported that Asian students perceive social environments and social perception of themselves as more shaming than personal or internalized shame (e.g., personal inferiority, self-blame) concerning mental health problems. Although these authors focused on shame towards mental health problems, it could be applied to sexual stigma. If so, then it could explain why guilt with an externalized object of negative evaluation would significantly impact mental health outcomes compared to ISS and shame with internalized objects of negative evaluation.

### 4.2. Limitations and Future Directions

This study had several limitations. Firstly, this is a single sample study at the intersection of sexuality and ethnicity, specifically Asian LGB individuals. This was performed to ensure decentralization from the majority White LGB population that tends to permeate studies in internalized sexual stigma. Also, we wanted to focus on a specific, underrepresented ethnic demographic within the LGBTQ+ community rather than overgeneralizing across multiple ethnicities that are living in the United States. However, this means that we were unable to determine if the outcomes that we found in this study were comparable to other ethnicities within the LGBTQ+ community or sexualities within the Asian American community, which would allow us to further unpack the role and relationships between sexual and ethnic identities and their related outcomes. Notwithstanding this limitation, the results of this study add to the literature about the relationship between ISS and mental health, especially for intersectional groups within the LGBTQ+ community. Future research could explore the nuanced differences in ISS between Asian nationality and/or ethnicity subgroups and also how ISS impacts LGB Asian Americans in other dimensions of mental health (e.g., the externalized spectrum) and wellbeing (including social and physical wellbeing).

Secondly, although this study broached into the intersectional impact of sexuality and ethnicity on ISS and mental health, we understand and acknowledge that there is still a multitude of other complex identities that intersect within and across the Asian LGB community. For example, country of birth, citizenship status, language spoken at home, and patterns across immigrant generations. Tikhonov and colleagues [69] have also shown that depression and anxiety are related to the identity integration of two cultures for ethnic minority American individuals, which could be extrapolated to multiracial LGB individuals. Although there are a variety of overlapping and intersecting identities, we were unable to analyze them in this paper due to the limited demographic question asked and sample size, which would have reduced the power for each intersectional analysis. This suggests that there may be more space for research to consider the impact of other demographic-based factors and multiracial LGB individuals.

A similar limitation relates to the aims of this study to explore the Intersectional experiences of Asian LGB individuals, but we acknowledge that (a) ‘Asian’ itself is a superordinate identity category and (b) the Asian LGB group arises from a diverse population (e.g., gender identity, age, and religion). As Asian is such a broad category, within this there will be other factors that might explain the correlations between ISS and mental health and low guilt or shame scores. For instance, if we had drawn from Asian cultures in which Islam was more dominant (adherents of Islam have higher levels of gay guilt and shame [70]), then some relationships between ISS, guilt, shame, and mental health might have been stronger or emerged. Additionally, shame can be operationalized into distinct experiences, such as external, internal, and reflected shame [68]. Further work is required to determine the role of ISS in explaining mental health outcomes against other groups, such as White LGB individuals or heterosexual Asians. For instance, reflected shame vs. internal shame vs. external shame and from varying social perspectives.

The mediation analyses in this study are correlational data, which are indicative of the relationships between ISS, guilt, shame, and mental health, but causality and directionality between these variables cannot be inferred. Therefore, this study could ideally be replicated with longitudinal data, although the process of collecting such data is complex and time-consuming. Finally, it is worth acknowledging that the analyses in this paper might be underpowered and thus could be vulnerable to a type II error. In the case where this were true, we could expect more of the findings presented in this paper to be statistically significant, particularly H1, which is a well-established effect in broader LGB samples (see [71] for a discussion).

### 4.3. Implications

This study contributes to the understandings of the impact of ISS on the mental health of Asian LGB individuals and the mediating roles of guilt and shame. These results may have implications for clinicians to consider integrating intersectionality into their practice when working with LGB clients from diverse ethnic backgrounds or other intersectional identities. It may also benefit emerging practitioners to understand how intersectionality in the LGBTQ+ community could exacerbate or mitigate ISS, or feelings of guilt or shame, which have a negative impact on mental health and wellbeing.

Additionally, previous public health research would suggest that these identities have relevance to, and impact upon, health-related behaviors and outcomes, although there is less existing research about how this appears in LGBTQ+ individuals.

## 5. Conclusions

This paper responds to a call for intersectional lenses to be applied to research in the field of stigma and health. In this paper, we applied this lens to psychological wellbeing of LGB Asian Americans, who constitute a group that are impacted by prejudices aimed at both their ethnic and sexual identities. The key findings from this paper are that, unexpectedly, a direct link between internalized sexual stigma and poorer mental health did not emerge for LGB Asian Americans (at least at the bivariate level and for this sample). However, the predicted minority stress effect emerged through the mediating variable of guilt. In addition, ISS was strongly related to the affective variables of shame and guilt.

This calls into question the assumption that sexual minority stress functions similarly for all ethnicity groups, while highlighting that this combination of identities do suffer from mental health disparities, relevant to the general population. We close with a call for continued research that explores intersectionality and culture-specific variables that may exacerbate or protect against poor mental health for individuals with multiple intersecting identities.

## Figures and Tables

**Figure 1 ijerph-21-00384-f001:**
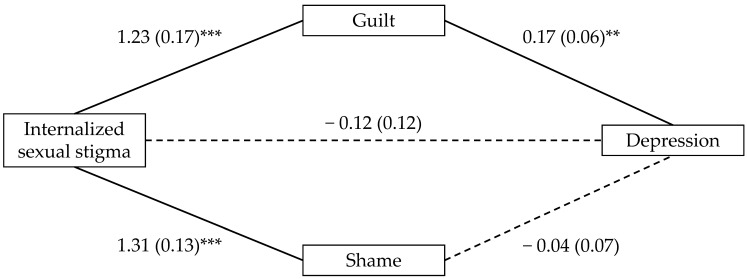
Indirect effects of internalized sexual stigma on depression. Notes. Values represent *B* (*SE*). *** *p* < 0.001, and ** *p* < 0.01. Solid lines reflect statistically significant paths; Dashed lines did not reach statistical significance.

**Figure 2 ijerph-21-00384-f002:**
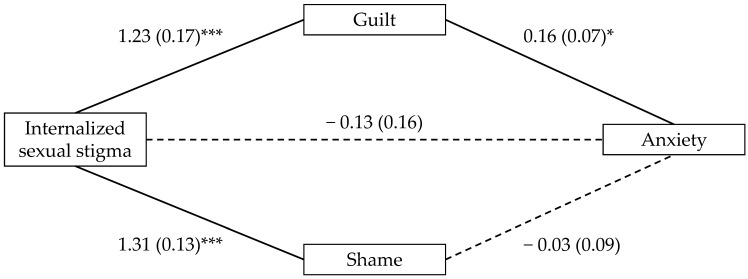
Indirect effects of internalized sexual stigma on anxiety. Notes. Values represent *B* (*SE*). *** *p* < 0.001, and * *p* < 0.05. Solid lines reflect statistically significant paths; Dashed lines did not reach statistical significance.

**Figure 3 ijerph-21-00384-f003:**
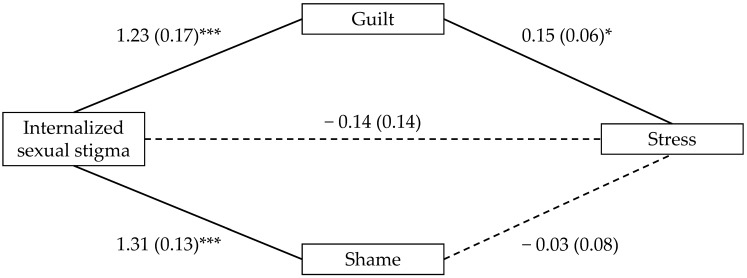
Indirect effects of internalized sexual stigma on stress. Notes. Values represent *B* (*SE*). *** *p* < 0.001, and * *p* < 0.05. Solid lines reflect statistically significant paths; Dashed lines did not reach statistical significance.

**Table 1 ijerph-21-00384-t001:** Descriptive statistics and correlations.

Variable	*M*	*SD*	Correlations				
			2	3	4	5	6
1. ISS	2.17	0.59	**0.52 ****	**0.65 ****	0.03	0.02	0.01
2. Guilt	2.10	1.39	–	**0.71 ****	**0.24 ****	**0.18 ***	**0.19 ***
3. Shame	2.12	1.20		–	0.11	0.09	0.08
4. Depression	1.39	0.67			–	**0.85 ****	**0.81 ****
5. Anxiety	1.39	0.87				–	**0.78 ****
6. Stress	2.21	0.76					–

Notes. ** *p* < 0.001, * *p* < 0.05 (2-tailed). Significant findings presented in boldface.

## Data Availability

The data presented in this study are available on request from the corresponding author.

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
