# Peer review of "Internalized Sexual Stigma and Mental Health Outcomes for Gay, Lesbian, and Bisexual Asian Americans: The Moderating Role of Guilt and Shame"

_ijerph, 2024, doi:10.3390/ijerph21040384_

Round 1

Reviewer 1 Report

Comments and Suggestions for Authors

The paper presents the results of a quantitative study assessing the relationship between internalised sexual stigma and various mental health outcomes, as well as the mediating roles of guilt and shame, among American-Asian LGB individuals. The introduction summarizes an extensive and diverse range of literature and presents a clear rationale for the present study. The study utilises a simple and well-designed quantitative survey which consists of multiple validated scales assessing relevant factors. The analyses and discussion are somewhat sound and my main qualm is regarding the results of the mediation analysis (which I will discuss below). Overall, a very well written paper and, despite the lack of statistically significant results, the paper was an enjoyable read and a worthwhile contribution to the field.

Line 35 – Avoid using the term ‘opposite’ to describe sex and/or gender as it implies a binary. https://apastyle.apa.org/style-grammar-guidelines/bias-free-language/gender

Line 52 – “could further elucidated” Should this be “could be further elucidated”?

Line 102 – “there are a range of factors differentially impact levels of ISS across LGB ethnic groups” This sentence reads a bit difficult. Does this mean factors that impact different levels of ISS? Could you please reword this sentence.

Line 163 – “feeling” add s

Line 179 – Just a suggestion, perhaps simply say 148 participants. “Non-heterosexual Asian people residing in the United States” was already stated in the previous sentence and repeating it makes the sentence long.

Line 184 – Missing parenthesis after n statistic.

Line 196 and Line 210 – add apostrophe to “participants level of…”

Line 202 – I’m confused what the Cronbach’s alpha is. Is this a typo?

Line 205 – Higher scores indicate higher degrees of guilt and shame… Was this also scored on a 5-point Likert scale?

Line 218 – Good use of an example utilising a Chinese population.

Line 321 – You state that guilt mediates the relationship between ISS and the mental health outcomes however there was no statistically significant relationship between the variables in the first place (based on the correlation analyses). It seems more likely that the results depict a relationship between guilt and depression, anxiety, and stress, independent of ISS. Furthermore, as mediation analyses can be seen as an extension of regression analyses, it would be interesting and helpful to also see this analysis included (i.e., include three regression analyses with only ISS and each of the mental health outcomes in the model without guilt and shame).

Line 327 – It might be helpful to re-state the percentages from the present study alongside the percentages from Terlizzi & Villarroel so that readers do not have to go back to the Results section to compare.

Line 388 – Indeed. Despite the analyses presenting statistically non-significant results, the study adds to the current literature and was a joy to read.

Line 392 – “acknoledge” Typo

Line 413 – “required to determine the compare…” Is this a typo? Should this be “required to compare”?

Line 423 – “result” add s

Line 426 – “intersectionalty” Typo

Line 429-431 – Consider separating this into two sentences. Add period between “outcomes although”

Line 437 – “it only emerges after accounting for third variables” As previously mentioned, it seems more likely that this relationship is more directly between guilt and the mental health outcomes and independent of ISS.

Comments on the Quality of English Language

Very well written and only minor typographical errors and corrections made. See previous comments.

Reviewer 2 Report

Comments and Suggestions for Authors

The current manuscript describes a mediation model where guilt and sham are mediators in the association between internalized sexual stigma and three mental health outcomes (depression, anxiety and stress). I believe that the overall manuscript is well written and describes the study in a straightforward fashion, however I have a few comments that I believe could improve the overall quality of your study.

1. Please consider to change the denomination of the study population in the title from “Asians” to “Asian Americans”. Because using only Asians could be misleading.

2. Please include some numerical results in you abstract.

3. Please describe with more detail some of the technical relevant information of your mediation models, e. g. did you use a parametric or non-parametric approach? Did you performed Sobell test or what type of statistical analysis was performed to test mediation. Additionally describe how did you test the prerequisites to test mediation (e. g. association for X to M and for M to Y). Also, if there were three independent models, please consider correcting p values for multiple comparisons.

4. Please consider discussing the potential impact of the sample size in the possibility of type 2 error. Mediation analysis is too demanding of sample size, and your n was about 200. Please discuss the potential implications

5. Please also consider the potential implications about how guilt and shame have small associations with mental health and larger associations with ISS. Could this mean that even High ISS is related to higher shame and guilty, but not with mental health problems?

6. Also consider that there is another dimension of mental health, this is, the externalized spectrum. I strongly suggest that you discuss how ISS could be potentially associated also with these disorders

Reviewer 3 Report

Comments and Suggestions for Authors

Dear authors,

It was my pleasure to review this manuscript which seeks to study the moderating role of guilt and shame on internalized sexual stigma and mental health outcomes of the Asian LGBT community.

With the sole objective of improving the quality of the manuscript, I will allow myself to make a series of comments.

1. Summary. The abstract should detail the study design.

2. Summary. Line 21. The acronym ISS has not been described previously.

3. Introduction. Point 2.1. I suggest not using acronyms in titles.

4. The introduction in general seems too long to me. I think it could be summarized a little to make it so heavy to read. Also, it is not necessary to divide the introduction into 4 sections. I think a single, well-structured section would be enough.

5. At the end of the introduction, the objective of the research must be explained clearly and explicitly. Reading section 1.4 I don't understand what the goal really is.

6. I think it is not necessary to explain the different hypotheses in section 1.4.

7. Methods. Section 2.1 should be the study design.

8. Section 2.1 should be called Population and Sample. The characteristics of the sample (line 182-191) belong to the descriptive or univariate study. I suggest moving this information to the results section.

9. The methodology does not explain what the inclusion and exclusion criteria were.

10. In the methodology section, a subsection called ethical considerations should be added, where these issues are addressed.

11. Results. Section 3.2 deals with descriptive and univariate analysis. It is not correct to include correlations at this point. Correlations are considered a bivariate study and do not fit in this section.

12. In correlations, the correlation coefficient and the p value must be expressed.

13. I think the conclusions should be rewritten, I don't see that those conclusions support the results of the study.

Thank you

Kind regards.
